# Increased risk of respiratory viral infections in elite athletes: A controlled study

**Maarit Valtonen**[1]*, **Wilma Grönroos**[2], **Raakel Luoto**[3], **Matti Waris**[4], **Matti Uhari**[5], **Olli J. Heinonen**[2], **Olli Ruuskanen**[3]

**1** Research Institute for Olympic Sports, Jyväskylä, Finland, **2** Paavo Nurmi Centre and Unit for Health and Physical Activity, University of Turku, Turku, Finland, **3** Department of Pediatrics and Adolescent Medicine, Turku University Hospital and University of Turku, Turku, Finland, **4** Institute of Biomedicine, University of Turku and Department of Clinical Virology, Turku University Hospital, Turku, Finland, **5** PEDEGO Research Unit, University of Oulu and Department of Pediatrics and Adolescents, Oulu University Hospital, Oulu, Finland

* maarit.valtonen@kihu.fi

**Data Availability Statement:** All relevant data are within the manuscript files.

**Funding:** This work was supported by Jenny and Antti Wihuri Foundation and Väinö and Laina Kivi Foundation. The funders had no role in study

## Abstract

### Background

Respiratory symptoms are commonly recognised in elite athletes. The occurrence, etiology and clinical presentation of the illnesses in athletes is unclear.

### Methods

We performed a prospective controlled study of respiratory viral infections in Team Finland during Nordic World Ski Championships 2019. There were 26 athletes and 36 staff members. Nasal swabs were taken at the onset of a symptom and on days 1, 7, and 13 during the follow-up of 14 days. Respiratory viruses were searched for by 3 different molecular multiplex tests. Fifty-two matched control subjects were studied in Finland during the same period.

### Results

Ten out of 26 (38%) athletes, 6 out of 36 (17%) staff, and 3 out of 52 (6%) control subjects experienced symptoms of respiratory infection (p = 0.0013). The relative risks for acquiring symptomatic infection were 6.7 (95% confidence interval [CI], 2.1–21.0) of athletes and 2.9 (95% CI, 0.84–10.0) of the staff as compared to the controls. Asymptomatic infections were identified in 8%, 22%, and 19%, respectively (p = 0.30). The etiology of respiratory infections was detected in 84% of the cases.

### Conclusion

The athletes had a 7-fold increase in the risk of illness compared to normally exercising control subjects.

design, data collection and analysis, decision to publish, or preparation of the manuscript.

**Competing interests:** The authors have declared that no competing interests exist.

## Introduction

A number of studies have suggested that continuous strenuous exercise induces a relative immunosuppression and elite athletes have enhanced susceptibility to acute and recurrent respiratory infections [1–4]. This phenomenon was originally described 30 years ago in marathon runners, of whom up to one third reported symptoms of an acute infectious episode after a race, the number being significantly higher than in runners who did not participate in the marathon [5]. More recently, during an intense period of competition, half of the elite cross-country skiers who participated reported becoming ill compared to one fifth of athletes who did not take part in the competition [6]. In most studies, the occurrence of illnesses has been based on self-reporting and the etiology of the infections has seldom been studied. Furthermore, normally exercising control subjects have usually not been included for comparison. Interestingly, the enhanced susceptibility of athletes to respiratory infections has recently been questioned and considered a misconception [7].

We previously reported in an uncontrolled study that 45% of athletes in Team Finland experienced symptoms of acute respiratory infection during 2018 PyeongChang Olympic Winter Games [4]. Importantly, viral aetiology of the infections was detected in 75% of the athletes.

The aims of this controlled study were to describe the occurrence, etiology and transmission of both symptomatic and asymptomatic respiratory viral infections in Team Finland during 2019 Nordic World Ski Championships. Fifteen respiratory viruses were searched for by molecular point-of-care test (POCT) on site. After the Championships, 16 respiratory viruses were tested for by 2 different multiplex PCR in laboratory-based testing.

## Material and methods

### Study planning and participants

This prospective observational controlled study was carried out during the Nordic World Ski Championships in Seefeld, Austria between Feb 18 and March 3, 2019. The study included Team Finland members who stayed in 2 hotels (31 athletes and 39 support staff members). One staff member was not included in the study because written consent could not be obtained. Two athletes were younger than 18 years of age and were ineligible. Three athletes and 2 staff members refused to participate (Fig 1).

The mean age ± SD of the athletes (n = 26) was 26 ± 4 years and that of the staff members (n = 36) 44 ± 9 years. Characteristics of 26 athletes are presented in Table 1. The team members were monitored for the entire duration of their stay in the hotels, starting from their arrival to the hotel and finishing with their departure from the hotel. On arrival, all team members were reported to be asymptomatic and healthy, except for 3 subjects with nasal congestion. The team members stayed in Seefeld for a median length of 14 days (with a range of 3–19 days).

Nasal swabs were taken from the team members on days 1, 7, and 13 with minor variation. All participants were instructed to immediately report respiratory symptoms to the team physician (M.V.). At the onset of symptoms, 2 nasal mucus specimens (1 from each nostril) were collected at a depth of 4–5 cm using flocked nasal swabs (503CS01, Copan Flock Technologies, Brescia, Italy). One specimen was used for POCT and the other was refrigerated in a dry storage tube. Follow-up specimens were taken on days 3, 6, and 9 of the illness with minor variation.

For every athlete, 2 normally exercising (<6 hours/week) controls were recruited from the students and staff of Turku University Hospital and University of Turku. They were matched

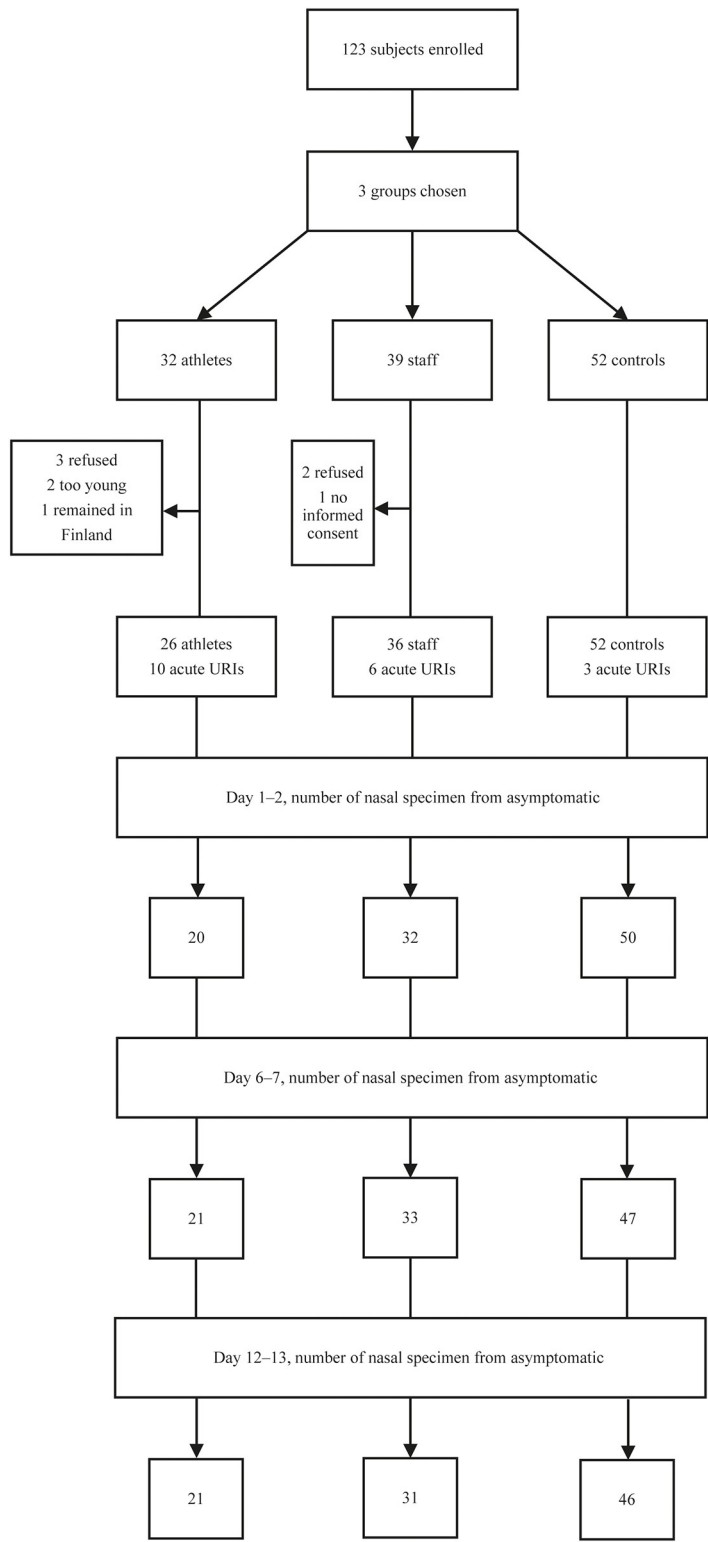

**Fig 1. Flow chart of the study.** [a]URI, upper respiratory infection.

Table 1. Characteristics of 26 athletes.

|  | Men | Women |
|---|---|---|
| **Cross-country skiing** | | |
| Number of athletes | 9 | 9 |
| Age, mean (range) | 27.4 (22–35) | 26.6 (19–31) |
| Athletes having children (N) | 2 | 0 |
| Smokers (N) | 0 | 0 |
| **Nordic Combined** | | |
| Number of athletes | 4 | 0 |
| Age, mean (range) | 22.5 (21–23) | |
| Athletes having children (N) | 0 | 0 |
| Smokers (N) | 0 | 0 |
| **Ski-jumping** | | |
| Number of athletes | 3 | 1 |
| Age, mean (range) | 22.7 (21–24) | 23 |
| Athletes having children (N) | 1 | 0 |
| Smokers (N) | 0 | 0 |

for age (± 2 years), sex, and the number of children younger than 5 years of age at home. The mean age was 27 ± 5 years. The control subjects were studied according to the protocol for Team Finland between Feb 24 and March 10. They filled out daily symptom cards and self-collected nasal swabs which were mailed to the laboratory [8].

All study-related activities were conducted according to Good Clinical Practice, which includes the provisions of the Declaration of Helsinki. The protocol was approved by the Ethics Committee of the Hospital District of Southwest Finland (ETMK 28/1801/2019). Informed written consent was obtained from 62 team members and 52 control subjects.

## Assessment of illness

Acute respiratory infection was defined as the acute onset of any of the following symptoms: sore throat, rhinorrhea, nasal congestion, and cough [4]. In addition, fever ($\geq$37.8˚C), hoarseness, and lethargy were recorded on a standardized form in the evenings on a 4-point severity scale (0 = absent, 1 = mild, 2 = moderate, and 3 = severe) [9]. The total symptom score for the first 5 days of illness was calculated.

## Microbiological studies

During the Games, an automated POCT, FilmArray Respiratory Panel 2 plus (BioFire; Salt Lake City, UT, USA), was used according to the manufacturer's instructions. The panel detects the following viruses: respiratory syncytial virus A and B, adenovirus, influenza A and B viruses, rhinovirus or enterovirus (without specifying which), parainfluenza type 1–4 viruses, human coronaviruses 229E, OC43, HKU1, and NL63, and human metapneumovirus. After a processing time of 60 minutes, the test reading was recorded.

Laboratory testing was carried out uniformly in Turku for the samples of the Team members and the subjects of the control group. Two PCR based tests were used: 1) Allplex Respiratory Panels 1–3 (Seegene, Seoul, South Korea) for respiratory syncytial virus A and B, adenovirus, influenza A and B viruses, rhinovirus, enteroviruses, parainfluenza type 1–4 viruses, human coronaviruses 229E, OC43, and NL63, human bocavirus, and human

metapneumovirus; 2) in-house triplex RT-PCR assay for respiratory syncytial virus, rhinoviruses, and enteroviruses [10]. The viral shedding was calculated as reported earlier [11].

## Statistical analysis

Data are presented as mean±SD and for non-normally distributed/skewed data median and interquartile range (IQR) were used. We followed 3 cohorts prospectively and calculated the relative risks and their 95% confidence intervals [CI] for the incidences of viral infections. When comparing all 3 cohorts, we calculated the chi square value for the 2 by 3 table.

## Results

### Subjects and specimens

A total of 114 subjects yielded 337 nasal swabs that were tested for viruses as follows: 28 specimens by FilmArray Panel on site, 336 by Allplex Panel, and 336 by triplex PCR panel in the laboratory.

### Presence of symptoms and clinical presentation

Sixteen out of 62 team members reported symptoms of respiratory infection during the study period and they were verified by the team physician: 10 athletes (38%) (1 female) and 6 (17%) staff members (Table 2 and Fig 2A). The median duration of the symptoms was 5.5 (IQR, 1.8–10.0) days and 6.0 (IQR, 2.5–8.8) days in the athletes and the staff, respectively. The symptoms were mostly mild, and the total median severity score was 4.0 (IQR, 1.8–18.8) for the athletes and 8.0 (IQR, 2.5–13.1) for the staff. Only 1 athlete could not compete on 1 race due to a respiratory infection. No team member suffered from a febrile illness.

Three out of 52 (6%) control subjects reported symptoms of respiratory infection during the 14-day study period (Table 2 and Fig 2B). In 2 cases the symptoms were verified by the research physician (R.L.) and 1 participant reported the symptoms by telephone. The median duration of symptoms was 8.0 days (IQR, 5.5–9.5) and the median severity score was 29.0 (IQR, 18.5–31.0). The relative risks for acquiring symptomatic infection were 6.7 (95% CI, 2.1–21.0) of athletes and 2.9 (95% CI, 0.84–10.0) of the staff as compared to the controls (Table 3).

### Etiology of the respiratory infections detected by molecular multiplex-POCT

On site, the etiology of respiratory infections was identified in 14 of 16 (88%) symptomatic cases. POCT detected the following viruses: rhinovirus (n = 6), respiratory syncytial virus B

**Table 2. Respiratory virus infections in team Finland and in controls in Turku.**

| | n | Virus detected | Symptomatic URI* | | | Asymptomatic URI | | Symptomatic of virus positive |
|---|---|---|---|---|---|---|---|---|
| | | | | Virus detection positive | | | | |
| | | n (%) | n (%) | n (%) | Ct (median) | n (%) | Ct (median) | n (%) |
| Athletes | 26 | 10 (38) | 10 (38) | 8 (80) | 23.27 | 2 (8) | 26.35 | 8 (80) |
| Staff | 36 | 14 (39) | 6 (17) | 6 (100) | 23.89 | 8 (22) | 31.66 | 6 (43) |
| Controls | 52 | 12 (23) | 3 (6) | 2 (67) | 17.65 | 10 (19) | 36.47 | 2 (17) |

*URI, upper respiratory infection

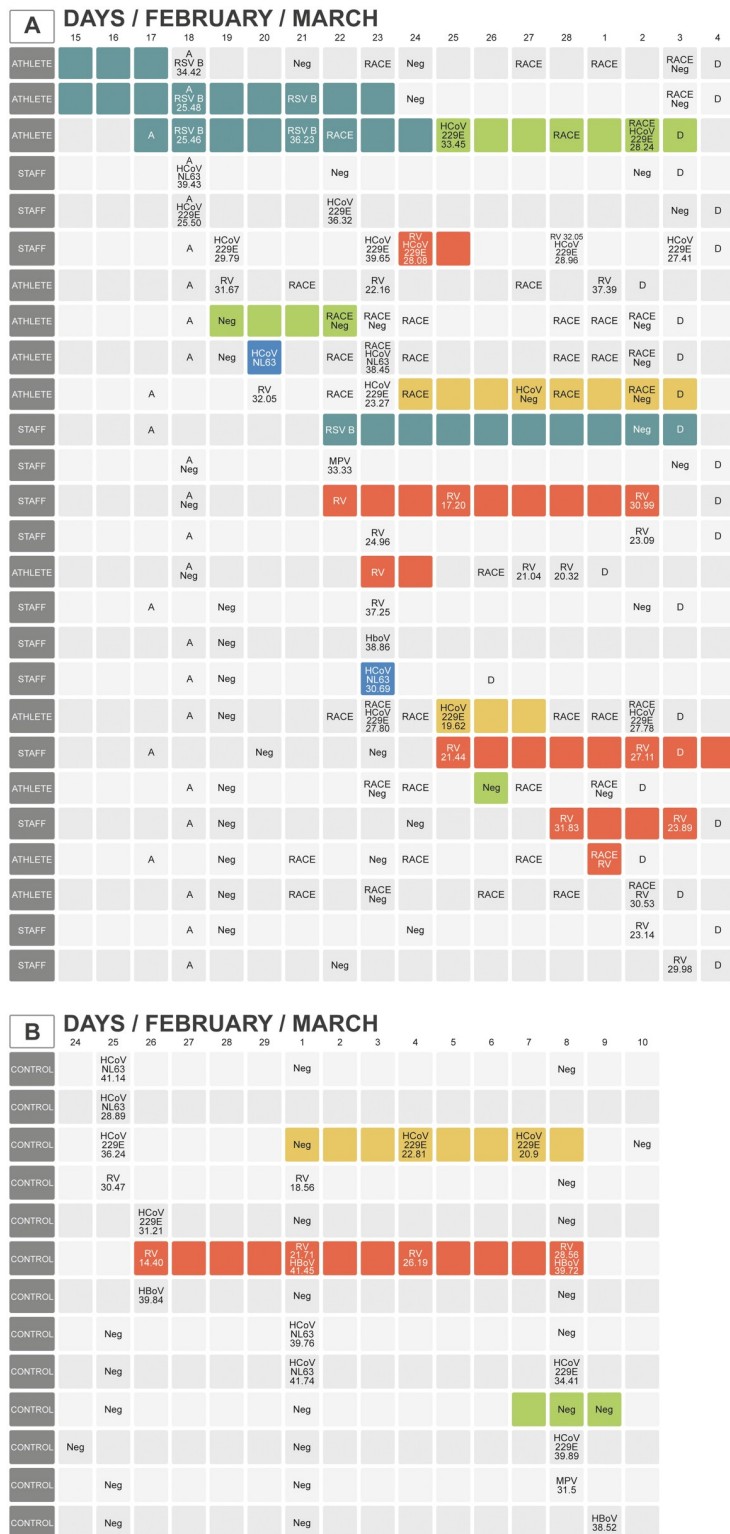

**Fig 2. Occurrence and etiology of viral infections.** The Fig 2A shows the transmission chain of symptomatic and asymptomatic infections in Team Finland. A = day of arrival at the hotel; D = departure from the hotel. The colored bars represent different viruses and the length of the bar indicates the duration of respiratory symptoms in days. RSV B, respiratory syncytial virus B; HCoV 229E, human coronavirus 229E; HCoV NL63, human coronavirus NL63; MPV, human metapneumovirus; RV, rhinovirus; HBoV, human bocavirus. The numbers in the boxes give Ct-values which

inversely reflect viral load, <27 reflects high viral load. Two viruses were only detected by POCT, so Ct-values are not given. The Fig 2B shows the corresponding information about control subjects.

(n = 4), coronavirus 229E (n = 2), and coronavirus NL63 (n = 2). The etiology of the infections was identified in 8 athletes (80%) and in 6 (100%) staff members (Tables 2 and 4).

## Etiology of the respiratory infections detected by 3 different multiplex-PCR-tests

Respiratory viruses were detected during the study period in 10 out of 26 (38%) athletes, in 14 out of 36 (39%) staff, and in 12 out of 52 (23%) controls (p = 0.20). Two athletes (8%) had an asymptomatic infection compared with 8 (22%) in staff and 10 (19%) in control subjects (p = 0.30) (Table 2).

Two multiplex PCR tests in the laboratory did not increase the etiologic yield of the symptomatic respiratory infections detected by POCT. However, 3 out of 8 negative follow-up tests in FilmArray were positive in the laboratory. Finally, 6 different respiratory viruses were identified in the Team: rhinovirus 13, coronavirus 229E 5, respiratory syncytial virus B 4, coronavirus NL63 3, metapneumovirus 1 and bocavirus 1 (Table 4 and Fig 2A). An in-house validated triplex PCR test detected 9 rhinoviruses not detected by the Allplex panel. One participant had

**Table 3. Relative risks of symptomatic and asymptomatic infections in the 3 cohorts.**

| Symptomatic infections in the 3 cohorts | | | Asymptomatic infections in the 3 cohorts | | |
|---|---|---|---|---|---|
| Cohort | Relative risk | 95% confidence interval | Cohort | Relative risk | 95% confidence interval |
| Controls | 1 | | Controls | 1 | |
| Staff | 2.9 | 0.8 to 10.0 | Staff | 1.2 | 0.5 to 2.6 |
| Athletes | 6.7 | 2.2 to 21.0 | Athletes | 0.4 | 0.1 to 1.4 |

Relative risk between athletes and staff 2.3, 95%, confidence interval 1.0 to 5.5. Relative risk between athletes and staff 0.35, 95% confidence interval 0.09 to 0.29

**Table 4. Symptomatic and asymptomatic respiratory viral infections in Team Finland and the controls.**

| | Symptomatic n | Virus n | Asymptomatic n | Virus n |
|---|---|---|---|---|
| Athletes | | | | |
| Cross-country skiing | 5 | Rhinovirus 2, RSV B* 2, negative 1 | 2 | Rhinovirus 2 |
| Nordic combined | 2 | Coronavirus 229E+rhinovirus 1, RSV B +coronavirus 229E 1 | 0 | |
| Ski jumping | 3 | Coronavirus 229E 1, coronavirus NL63 1, negative 1 | 0 | |
| Staff members | | | | |
| Coaches | 2 | RSV B 1, rhinovirus 1 | 0 | |
| General managers and communications | 0 | | 2 | Coronavirus 229E 1, metapneumovirus 1 |
| Ski technicians | 4 | Rhinovirus 2, rhinovirus+coronavirus 229E 1, coronavirus NL63 1 | 4 | Rhinovirus 3, bocavirus 1 |
| Medical personnel | 0 | | 2 | Rhinovirus 1, coronavirus NL63 1 |
| Controls | 3 | Coronavirus 229E 1, rhinovirus +bocavirus 1, negative 1 | 10 | Coronavirus 229E 3, coronavirus NL63 3, metapneumovirus 1, bocavirus 2, rhinovirus 1 |

*RSV B, respiratory syncytial virus B

a dual infection, rhinovirus and coronavirus 229E (Fig 2A, line 6), and two had serial infections, 1 respiratory syncytial virus B and coronavirus 229E (Fig 2A, line 3) and the other rhinovirus and coronavirus 229E (Fig 2A, line 10). The median PCR cycle threshold (Ct) values for the 3 study groups were similar. In 6 athletes and in 6 staff members the Ct values were <27, indicating high viral load (Fig 2A). The mean duration ± SD of viral shedding was 7.8 ± 3.1 days (n = 9) in the athletes and 6.6 ± 3.0 days (n = 11) in the staff.

Five different viruses were detected in the control subjects. In 1 symptomatic subject, dual rhinovirus and bocavirus infections were detected on 2 occasions (Fig 2B, line 6). In 8 cases the virus was detected only once out of the 3 consecutive nasal swabs (Fig 2B). Three nasal specimens at 1-week intervals revealed viruses in 10%, 6%, and 9% of asymptomatic control subjects (mean occurrence 8%, total 19%) and in 9%, 18%, and 14% of asymptomatic staff members (mean occurrence 14%, total 22%) (Fig 1). The athletes seemed to have fewer asymptomatic viral infections (10%, 5%, 11%, mean occurrence 9%, total 8%), but the difference was not significant.

Only 2 Team members were treated with antibiotics due to suspected bacterial infection.

## Competition and respiratory infections

Five virus-negative athletes became virus-positive after their competition; 3 with symptomatic infection and 2 with asymptomatic infection. Six athletes competed when suffering from a viral infection, 2 with mild symptoms, 4 without symptoms, and 1 of them developed respiratory symptoms after the competition. One athlete with an earlier respiratory syncytial virus B infection acquired a coronavirus 229E infection after the competition (Fig 2A, line 3). In no symptomatic participant enhancement of symptoms could be verified after the competition.

## Time-course of infections

On arrival 7 subjects were virus-positive. Seventeen team members (8 athletes) arrived as virus-negative and were infected during their stay in Seefeld. Nine of them were infected during the first 5 days of their stay. Upon departure 15 of 24 team members, who were infected at some point, were still virus-positive (Fig 2A).

## Discussion

The main findings of this prospective controlled study are 3-fold. First, 38% of the athletes suffered from symptomatic respiratory infection during the median stay of 2 weeks in Seefeld. The athletes had a 7-fold increase in the risk of illness compared with normally exercising control subjects and a 2-fold risk compared with the staff. Second, with 3 different multiplex PCR tests, the probable etiology of infections could be detected in 80% of the athletes. Rhinoviruses and coronaviruses were the most common causative agents. However, several other respiratory viruses circulated within the Team. Third, asymptomatic respiratory virus infections were detected in 8% of the athletes but in 22% and 19% of the controls and the staff, respectively. In some athletes, competing was associated with the development of symptomatic and asymptomatic viral infections.

Our observations (38%) and those of 2 earlies studies confirm that athletes have significantly enhanced susceptibility to respiratory viral infections during major winter sport games [4, 12]. We reported recently in an uncontrolled study that 45% of the athletes of Team Finland experienced symptoms of respiratory infection during the 3-week 2018 Olympic Winter Games [4]. During the 8-11-day Tour de Ski or 10 days immediately after the event, 48% of Norwegian cross-country skiers reported symptoms of respiratory viral infection [12]. The key

question is what the mechanisms behind the 7-fold risk to respiratory viral infections in athletes during a major winter sport event are and how can we influence them.

It is commonly reported, but not well documented, that heavy exercise-induced immunosuppression, mental stress, nutritional restrictions, air travel, sleep disturbance, human crowding, housing with other athletes, low temperature with low humidity, and competition all increase the risk of respiratory virus infection, especially during the winter time when many viruses are prevalent [3]. It is of note that the athletes had a 2-fold risk for illness when compared with the staff, who share many risk factors with the athletes. On the other hand, the staff had a 3-fold increased risk compared with the control subjects, suggesting that travelling and crowding are also important risk factors [13]. It is tempting to link competition with heavy physical and mental stress and enhanced susceptibility to respiratory viral infections. This speculation is supported by our observations from the 2018 Olympic Games where 71% of 14 cross-country skiers had acute respiratory illness compared to 26% of 27 ski technicians [4]. In the present study, asymptomatic viral infection developed into symptomatic infection in 4 athletes after the race.

We detected the etiology of acute respiratory symptoms in 80% of the athletes in Seefeld and in 75% in PyeongChang Olympic Games [4]. These observations clearly show that athletes' acute upper respiratory symptoms are caused by viral infections and "non-infectious airway inflammation or nonspecific upper respiratory symptoms' may not exist [2]. Multiplex PCR tests for 17 viruses detected 6 different respiratory virus infections, which were all detected at the same time on days 4–5 after arrival. The great number of viruses reflects the numerous possible sources of infections outside the team [14]. Although the athletes have enhanced susceptibility to respiratory viral infections, the clinical symptoms were mostly mild, and the viral loads were mostly low. Four athletes had a paucisymptomatic (1 symptom only) viral infection as is commonly seen, for example, in influenza [15]. Furthermore, short viral shedding times were recorded. These observations do not support clinically significant immunosuppression in elite athletes [7].

This is the first study in which respiratory viruses were systematically searched for also in asymptomatic subjects of a sports team during major Games. Three nasal specimens at 1-week intervals revealed 5 different viruses in 19% of the asymptomatic control subjects and in 22% of the staff (Fig 2A and 2B). This observation is in agreement with the 1-year follow-up study in which a respiratory virus was detected in adults in 16% of the weeks and 45% of those viral detection episodes were asymptomatic [16]. Unexpectedly, the athletes with the marked prevalence of viruses had few asymptomatic viral infections (8%). Only 2 out of 10 virus detections were asymptomatic (Table 4). This observation supports the hypothesis that the physical and mental stress of competing may activate respiratory symptoms in athletes with asymptomatic virus infection (Fig 2A). The clinical significance of respiratory viruses in asymptomatic subjects is debatable. Respiratory RNA viruses (with the exception of coxsackieviruses) do not cause chronic asymptomatic infections in healthy humans and it is well established that a great share of respiratory virus infections are asymptomatic [17]. In our study, one-third of rhinovirus and coronavirus infections were asymptomatic. How much an asymptomatic person with non-SARS-CoV-2 infection can transmit the infection and how often asymptomatic infections develop into symptomatic infections is unclear.

Our observation suggests that molecular POCT is accurate in the hands of trained team physicians. The need for expensive comprehensive POCT in sport events can be questioned because a specific therapeutic intervention is only available for influenza. We think that through prompt recognition of symptoms and early specimen collection with early precise viral diagnosis, patients can be quickly isolated and cohorted (both symptomatic and asymptomatic) thereby mitigating the risk of transmission as was done in Team Finland. In this way,

testing may translate to athlete benefit [18]. Importantly, only 2 Team members were treated with antibiotics and none with oseltamivir.

Seventeen of 24 subjects with viral infection were virus-negative when they arrived, and of these, 10 developed a symptomatic infection and 7 an asymptomatic infection. Interestingly, 59% of the infections developed on site during the first 5 days suggesting transmission of infection during travelling. In the middle of the games 25% of the Team were virus positive (Fig 2A).

The strengths of this prospective controlled study are the 3 established laboratory PCR techniques, the utilization of POCT on site and the objective symptom data collection by a team physician. There are, however, some limitations to our study. The Team consisted of only 26 athletes, 10 of them developed a symptomatic infection with 4 different viruses. Thus, the number of index cases were small, and some observations should be considered with care. The prevailing viruses in the community in Austria may have contributed to our observations. However, during the study period an influenza epidemic was occurring in Austria, but no cases of influenza were detected in team Finland. All the viruses detected in the Team (in Austria) were also prevalent in Turku, Finland. The team's nasal samples were collected professionally, but control subjects self-collected their samples. We could not dissect the mechanisms behind the enhanced susceptibility to infections.

In conclusion, during the Winter Sport Championship Games, the athletes had a 7-fold increase in the risk of acute respiratory tract infections. Most infections in athletes were symptomatic but the symptoms were mild. Asymptomatic infections were commonly detected in staff and control subjects, potentially transmitting the infections. The etiology of the infections was identified in the majority of cases.

## Acknowledgments

We thank the Olympic Committee of Finland and the Finnish Ski Federation for their support. We thank BioMerieux, Finland, for supplying the FilmArray System. We are grateful to Minna Hyppönen and Minna Pirttinen for technical assistance in the laboratory.

## Author Contributions

**Data curation:** Maarit Valtonen, Raakel Luoto, Matti Waris.

**Formal analysis:** Matti Uhari.

**Funding acquisition:** Olli Ruuskanen.

**Investigation:** Maarit Valtonen, Wilma Grönroos, Raakel Luoto.

**Methodology:** Maarit Valtonen, Raakel Luoto, Matti Waris, Olli J. Heinonen, Olli Ruuskanen.

**Project administration:** Maarit Valtonen.

**Resources:** Olli Ruuskanen.

**Supervision:** Olli J. Heinonen, Olli Ruuskanen.

**Visualization:** Olli Ruuskanen.

**Writing – original draft:** Maarit Valtonen, Olli Ruuskanen.

**Writing – review & editing:** Maarit Valtonen, Wilma Grönroos, Raakel Luoto, Matti Waris, Olli J. Heinonen, Olli Ruuskanen.

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
