## [Decision Letter · Decision Letter 0]

8 Mar 2021

PONE-D-20-25451

Increased risk of respiratory viral infections in elite athletes a controlled study

PLOS ONE

Dear Dr. Valtonen,

Thank you for submitting your manuscript to PLOS ONE. After careful consideration, we feel that it has merit but does not fully meet PLOS ONE’s publication criteria as it currently stands. Therefore, we invite you to submit a revised version of the manuscript that addresses the points raised during the review process.

We look forward to receiving your revised manuscript.

Kind regards,

Davor Plavec, MD, MSc, PhD, Prof.

Academic Editor

PLOS ONE

Additional Editor Comments:

Dear Authors,

the theme of the manuscript is really interesting, but the work has some problems which are not enough discussed. The first one is a small sample size with a small number of index cases that should be discussed in more detail and the second one is the control group that was assessed in Finland where the epidemiological situation could be significantly different from the one in Austria. It is difficult to understand the need to enclose the data from a control group in Finland in this respect. Also the methodology used for swabs is significantly different and also the used diagnostic panels were different. There is no detailed explanation how the use of these different protocols and methods could influence the obtained results. For instance what is the comparative sensitivity and specificity of different diagnostic panels, etc. Please revise your manuscript in these respects.

Journal Requirements:

2. Please state whether the baseline demographic characteristics of the study populations were recorded. If so, please provide a table summarising these.

3. Please provide a summary of the current literature within the Introduction section, putting your current study into context such that it may be understood by someone outside of this area of expertise. Please see our submission guidelines for further details http://journals.plos.org/plosone/s/submission-guidelines#loc-parts-of-a-submission.

"We thank the Olympic Committee of Finland and the Finnish Ski Federation for their support."

"This work was supported by Jenny and Antti Wihuri Foundation and Väinö and Laina Kivi Foundation. The funders had no role in study design, data collection and analysis, decision to publish, or preparation of the manuscript."

Reviewers' comments:

Reviewer's Responses to Questions

**Comments to the Author**

1. Is the manuscript technically sound, and do the data support the conclusions?

Reviewer #1: Yes

Reviewer #2: Partly

2. Has the statistical analysis been performed appropriately and rigorously? 

Reviewer #1: Yes

Reviewer #2: Yes

3. Have the authors made all data underlying the findings in their manuscript fully available?

Reviewer #1: Yes

Reviewer #2: Yes

4. Is the manuscript presented in an intelligible fashion and written in standard English?

Reviewer #1: Yes

Reviewer #2: Yes

5. Review Comments to the Author

Reviewer #1: It was a pleasure reading this manuscript. The manuscript demonstrates a technically sound piece of scientific research with data that supports the conclusions. The statistical analysis been performed appropriately and rigorously.

Looking forward to reading future research from the authors.

Reviewer #2: In the manuscript, PONE-D-20-25451 "Increased risk of respiratory viral infections in elite athletes a controlled study" authors investigate the occurrece of respiratory viralinfections in Team Finland during Nordic World Ski Championships 2019. in 26 athletes and 36 staff members as well as in 52 matched cintrols.

- The samples of nasal swabs were taken from the team members in completely different conditions (in Austria) compared to the control group in which the samples were collected in Finland.

- The next limitation of this study thet samples of control subjects self-controlled unlike team's nasal samples were collected professionally.

- The number of subjects and virus positive cases were small which the authors cite as the limitaion of the study.

6. PLOS authors have the option to publish the peer review history of their article (what does this mean?). If published, this will include your full peer review and any attached files.

Reviewer #1: No

Reviewer #2: No

---

## [Author Response · Author response to Decision Letter 0]

30 Mar 2021

The comments of the Academic Editor and our responses:

1) The first one is a small sample size with a small number of index cases that should be discussed in more detail 

> We have added the following sentences to the discussion in the paragraph of limitations: The Team consisted of only 26 athletes, 10 of them developed a symptomatic infection with 4 different viruses. Thus, the number of index cases were small, and some observations should be considered with care. 

2)...the control group that was assessed in Finland where the epidemiological situation could be significantly different from the one in Austria. 

> Dr. Monika Redlberger-Fritz, Center for Virology, Medical University Vienna, informed us that during the Winter Games there were influenza epidemic as well as minor outbreaks of RSV and rhinoviruses occurring in Austria.

> We have added the following sentences to the discussion in the paragraph of limitations:

The prevailing viruses in the community in Austria may have contributed to our observations. During the study period, an influenza epidemic as well as outbreaks of RSV and rhinovirus were occurring in Austria. However, no cases of influenza was detected in Team Finland. All 6 respiratory viruses detected in the Team (in Austria) were also prevalent in Turku, Finland.

3) It is difficult to understand the need to enclose the data from a control group in Finland in this respect. 

> Most of earlier studies on the topic have been without a control group. We think it was important to include an age and sex matched, normally exercising control group in Finland. That group lacked all the risk factors athletes were predisposed when participating the Games: heavy physical exercise (700-900 h/year), psychological distress, travelling, shared housing, mass gathering, competitions. We also want to stress that we included the supportive staff of the Team as a comparison group on site sharing many of the risk factors with the athletes.

4) Also, the methodology used for swabs is significantly different and also the used diagnostic panels were different. There is no detailed explanation how the use of these different protocols and methods could influence the obtained results. For instance what is the comparative sensitivity and specificity of different diagnostic panels, etc.

> We admit that we were not careful when writing the chapter Microbiological studies.

Multiplex PCR (Allplex) and in-house triplex PCR were carried out for specimens of all participants. POCT was just an additional test, which did not change the final virological observations. Nasal sampling using Copan swabs were taken on the same way from all participants. We and others have shown in several studies (recently also for COVID-19) that the self-collection results are comparable to those of professional collection. With than in mind, we think that there is no bias in sample collection and viral diagnostics.

We edited the corresponding paragraph as follows:

Laboratory testing was carried out uniformly in Turku for the samples of all Team members and the subjects of the control group. Two PCR based tests were used: 1) Allplex Respiratory Panels 1-3 (Seegene, Seoul, South Korea) for respiratory syncytial virus A and B, adenovirus, influenza A and B viruses, rhinovirus, enteroviruses, parainfluenza type 1-4 viruses, human coronaviruses 229E, OC43, and NL63, human bocavirus, and human metapneumovirus; 2) in-house triplex RT-PCR assay for respiratory syncytial virus, rhinoviruses, and enteroviruses [8].

Journal Requirements

> We have tried to be more careful in the resubmitted manuscript. 

2. Please state whether the baseline demographic characteristics of the study populations were recorded. If so, please provide a table summarising these.

> A table summarizing demographic characteristics have been added as Table 1.

3. Please provide a summary of the current literature within the Introduction section, putting your current study into context such that it may be understood by someone outside of this area of expertise.

> We have added the following paragraphs within the introduction:

This phenomenon was originally described 30 years ago in marathon runners, of whom up to one third reported symptoms of an acute infectious episode after a race, the number being significantly higher than in runners who did not participate in the marathon [5]. More recently, during an intense period of competition, half of the elite cross-country skiers who participated reported becoming ill compared to one fifth of athletes who did not take part in the competition [6].

We previously reported in an uncontrolled study that 45% of athletes in Team Finland experienced symptoms of acute respiratory infection during 2018 PyeongChang Olympic Winter Games. Importantly, viral etiology of the infections was detected in 75% of the athletes [4].

"We thank the Olympic Committee of Finland and the Finnish Ski Federation for their support."

We note that you have provided funding information that is not currently declared in your Funding Statement. However, funding information should not appear in the Acknowledgments section or other areas of your manuscript. 

>The support of the Olympic Committee of Finland and the Finnish Ski Federation was not financial. So, we have changed the sentence as follows:

We thank the Olympic Committee of Finland and the Finnish Ski Federation for their organizational support.

We also added acknowledgement: We are grateful to Minna Hyppönen and Minna Pirttinen for technical assistance in the laboratory.

The funding statement has been removed from the main text:

This work was supported by Jenny and Antti Wihuri Foundation and Väinö and Laina Kivi Foundation. 

Reviewer 1

> We thank the reviewer for his comments.

Reviewer 2

- The samples of nasal swabs were taken from the team members in completely different conditions (in Austria) compared to the control group in which the samples were collected in Finland.

- The next limitation of this study thet samples of control subjects self-controlled unlike team's nasal samples were collected professionally.

> We response as stated earlier:

Multiplex PCR (Allplex) and in-house triplex PCR were carried out for specimens of all participants. POCT was just an additional test, which did not change the final virological observations. Nasal sampling using Copan swabs were taken on the same way from all participants. We and others have shown in several studies (recently also for COVID-19) that the self-collection results are comparable to those of professional collection. With than in mind, we think that there is no bias in sample collection and viral diagnostics.

- The number of subjects and virus positive cases were small which the authors cite as the limitaion of the study.

> We have added the following sentences to the discussion in the paragraph of limitations: The Team consisted of only 26 athletes, 10 of them developed a symptomatic infection with 4 different viruses. Thus, the number of index cases were small, and some observations should be considered with care.

---

## [Decision Letter · Decision Letter 1]

16 Apr 2021

Increased risk of respiratory viral infections in elite athletes: a controlled study

PONE-D-20-25451R1

Dear Dr. Valtonen,

We’re pleased to inform you that your manuscript has been judged scientifically suitable for publication and will be formally accepted for publication once it meets all outstanding technical requirements.

Kind regards,

Davor Plavec, MD, MSc, PhD, Prof.

Academic Editor

PLOS ONE

Additional Editor Comments (optional):

The manuscript is acceptable for publication in it's current form.

Reviewers' comments:

Reviewer's Responses to Questions

**Comments to the Author**

1. If the authors have adequately addressed your comments raised in a previous round of review and you feel that this manuscript is now acceptable for publication, you may indicate that here to bypass the “Comments to the Author” section, enter your conflict of interest statement in the “Confidential to Editor” section, and submit your "Accept" recommendation.

Reviewer #2: All comments have been addressed

2. Is the manuscript technically sound, and do the data support the conclusions?

Reviewer #2: Yes

3. Has the statistical analysis been performed appropriately and rigorously? 

Reviewer #2: Yes

4. Have the authors made all data underlying the findings in their manuscript fully available?

Reviewer #2: Yes

5. Is the manuscript presented in an intelligible fashion and written in standard English?

Reviewer #2: Yes

6. Review Comments to the Author

Reviewer #2: The authors have adequately addressed all comments raised in a previous round of review and I feel that this manuscript is now acceptable for publication,

7. PLOS authors have the option to publish the peer review history of their article (what does this mean?). If published, this will include your full peer review and any attached files.

Reviewer #2: No

---

## [Editor Report · Acceptance letter]

22 Apr 2021

PONE-D-20-25451R1 

Increased risk of respiratory viral infections in elite athletes:a controlled study 

Dear Dr. Valtonen:

I'm pleased to inform you that your manuscript has been deemed suitable for publication in PLOS ONE. Congratulations! Your manuscript is now with our production department. 

Kind regards, 

on behalf of

Dr. Davor Plavec 

Academic Editor

PLOS ONE